# Obesity and Abdominal Obesity in Indian Population: Findings from a Nationally Representative Study of 698,286 Participants

Rajat Das Gupta [1,*], Nowrin Tamanna [1], Nazeeba Siddika [2,3], Shams Shabab Haider [4], Ehsanul Hoque Apu [3,5,6] and Mohammad Rifat Haider [7]

1 Department of Epidemiology and Biostatistics, Arnold School of Public Health, University of South Carolina, Columbia, SC 29028, USA

2 Department of Epidemiology and Biostatistics, College of Human Medicine, Michigan State University, East Lansing, MI 48824, USA

3 Centre for International Public Health and Environmental Research, Bangladesh (CIPHER,B), Dhaka 1207, Bangladesh

4 Johns Hopkins Bloomberg School of Public Health, 615 N Wolfe St., Baltimore, MD 21205, USA

5 Department of Biomedical Engineering, Institute for Quantitative Health Science and Engineering (IQ), Michigan State University, East Lansing, MI 48824, USA

6 Research Unit of Medical Imaging, Physics and Technology, Faculty of Medicine, University of Oulu, 90220 Oulu, Finland

7 Department of Health Policy and Management, College of Public Health, University of Georgia, Athens, GA 30602, USA

* Correspondence: rajatdas@email.sc.edu

**Abstract:** This study aims to determine and compare the prevalence and correlates of obesity and abdominal obesity in India among participants aged 18–54 years. Data were acquired from the nationally representative National Family Health Survey 2019–21. Age and sex standardized descriptive analyses were conducted to determine the prevalence of obesity and abdominal obesity, and multivariable multilevel logistic regression was performed to identify the factors associated with these conditions. Gender-specific analyses were also conducted. The sample weight was adjusted throughout. The final sample size for this study was 698,286. The prevalence of obesity and abdominal obesity was 13.85% and 57.71%, respectively. Older age, being female, increased educational status and increased wealth index, being married at any point, and residing in an urban area all increased the odds of both obesity and abdominal obesity. Being a resident of the North zone and having a current alcohol intake increased the odds of abdominal obesity. On the other hand, being a resident of the South zone of India increased the odds of obesity. Targeting these high-risk groups can be a strategy for public health promotion programs.

**Keywords:** obesity; abdominal; risk factors; prevalence

## 1. Introduction

Obesity, defined as abnormal or excessive fat accumulation, which may impair health, is measured by body mass index (BMI) $\geq 30$ kg/m$^2$ [1]. In the Asian population, the BMI cut-off of obesity is lower ($\geq$27.5 kg/m$^2$) [2]. Anatomically, adipose tissue is unequally distributed in varying proportions in the human body [3]. When the distribution of adipose tissue is abnormally high surrounding the internal viscera of the abdomen, it is referred to as abdominal or central obesity [4]. Both obesity and abdominal obesity are associated with increased risk of morbidities such as systematic inflammation, insulin resistance, and lipid abnormalities, leading to various noncommunicable diseases, including cerebrovascular and cardiovascular (CVD) diseases and cancer [5–7], as well as mortality. Additionally, abdominal obesity has a heightened risk of lifetime disability and poor long-term survival, regardless of BMI category, compared to people with no abdominal obesity [8,9].

The prevalence of abdominal obesity is increasing rapidly due to increased sedentary lifestyles, physical inactivity, and unhealthy diet [10]. Approximately 13% of the world's adult population was suffering from obesity in 2016. It has been estimated that nearly 41.5% of the world's adult population has abdominal obesity [10]. Yet, it remains one of the least addressed health issues, and is a major public health concern in both upper-income and low-and-middle-income countries (LMICs).

In the Asian subcontinent, obesity has become quite common, particularly in several South Asian nations [11]. Genetically, among all races, Asian people tend to store more fat around the abdomen [11]. With recent economic progress and urbanization, traditional healthy diets have been substituted by Western-influenced diets rich in saturated fats, refined sugar, and preservatives. Excessive consumption of white rice and a lack of physical activity may also contribute to the deposition of abdominal fat among South Asian people [12,13]. India, the largest country in South Asia, with 1.2 billion people, is believed to have an estimated 350 million people with obesity [14].

Although the prevalence of abdominal obesity in India's general population is unknown, several studies indicate a worrisome upward trend [14,15]. These studies were conducted within some specific subpopulation(s), which do not represent the entire population of India. To address this gap, we aim to estimate the prevalence and correlates of abdominal obesity (aged 18–54 years) and compare it with the prevalence and factors related to obesity using a nationally representative sample of the Indian population obtained from the National Family Health Survey 2019–21 (NFHS-5).

## 2. Materials and Methods

### 2.1. Data

A secondary analysis of the NFHS-5 2019–21 was conducted. NFHS-5 was a nationally representative survey implemented in India. Detailed reports of the methodology, sample size calculation, and preliminary findings have been published elsewhere [16,17]. Stratified sampling strategy was followed for data collection. Data were collected in two stages. First, primary sampling units (PSUs) were selected from the rural area and census enumeration blocks (CEBs) were selected from the urban area based on probability proportionate to size. At the second stage, an equal number of households ($n = 22$) were selected randomly from each PSU. From the selected households, all the eligible men and women were included for interview [16,17].

### 2.2. Data Collection and Measurement

Four questionnaires were used for data collection (household, woman, man, and biomarker). Questionnaires were pretested and validated according to the local context. Participants' height and weight were measured using two Seca devices (213 stadiometers and 874 digital scales). Gulick tapes were used for measuring waist and hip circumferences [16].

### 2.3. Outcome Variable

Both obesity and abdominal obesity were outcomes of interest for this study, which were dichotomized into (i) "yes" (presence of obesity/abdominal obesity), (ii) "no" (absence of obesity/abdominal obesity). Abdominal obesity was defined as a waist–hip ratio of >0.90 for males and >0.85 for females, as determined by the WHO [18]. An Asia-specific cut-off was used for BMI categorization: underweight ($<18.5$ kg/m$^2$); normal weight ($18.5$ kg/m$^2$–$<23.0$ kg/m$^2$); overweight ($23.0$ kg/m$^2$–$<27.5$ kg/m$^2$); obesity ($\geq 27.5$ kg/m$^2$) [2].

### 2.4. Explanatory Variables

We included participant's age, gender, highest educational status, marital status, wealth index, zone and place of residence, current smoker/tobacco use, and current alcohol consumption as explanatory variables.

Age was categorized into 18–24, 25–34, 35–44, and 45–54 years. Gender was categorized into male and female. Academic level was categorized into no formal schooling, up

to primary, up to secondary, and college and higher. NDHS-5 collected information on household assets, including construction materials, water sources, health and sanitation facilities, and electricity usage. Since DHS program did not collect information on household income, the wealth index, which is a composite measure of a household's cumulative living standard, was used to represent the economic condition of the households. This information was used to compute a wealth index using principal component analysis [16]. After computing the wealth index, it was categorized into poorest, poorer, middle, richer, and richest. Marital status was categorized into never married, currently married, and others (divorced/widowed/separated/don't know). The residence zone was categorized according to the current administrative zones of India: North, North-East, Central, East, West, and South [19]. The place of residence (location) was divided into urban and rural areas. Both current smoker/tobacco use and current alcohol consumption were categorized into either "yes" or "no".

### 2.5. Statistical Analysis

For descriptive analyses of our study population, we calculated both unweighted and weighted percentages (mean and standard deviation) of the categorical variables. The sampling weight of NHFS-5 was used. Bivariate analysis was conducted to explore the proportion of obesity and abdominal obesity according to the subcategories of the explanatory variables. Although the male–female ratio in India is 1.02:1.00, in this study, the male female ratio was 1:8 [20]. So, age and sex were standardized while estimating the prevalence of obesity and abdominal obesity in the general population.

Multilevel logistic regression was performed to identify the factors associated with both obesity and abdominal obesity, adjusted for the cluster effect and the complex hierarchical structure of DHS data [21]. Both crude and adjusted odds ratio (COR and AOR respectively) with their 95% confidence interval (CI) were reported. We further stratified the analyses by gender.

We used Stata Version 17.0 for all statistical analyses [22].

### 2.6. Ethical Considerations

The study protocol of NFHS-5 was approved by the institutional review board of the International Institute for Population Sciences (IIPS) and ICF [16]. The U.S. Centers for Disease Control and Prevention (CDC) also reviewed the protocol. Prior to data collection, informed consent was taken from the study participants. We received approval from DHS program for using the NFHS-5 data for this study.

### 3. Results

Table 1 presents the background characteristics of the participants as well as the prevalence of obesity and abdominal obesity by background characteristics. Overall, 698,286 participants were included in the study. Most of them were aged between 25 to 34 years (32.67%), female (87.57%), educated up to secondary level (46.95%), and married (78.20%). A total of 67.98% of the participants were from rural areas. Around 10.71% of the participants were current smokers/tobacco users, and approximately 3.60% of the participants were current alcohol consumers during the time of the survey.

**Table 1.** Prevalence of obesity and abdominal obesity by background characteristics (N = 698,286).

| Variable | Unweighted Frequency n (%) | Prevalence of Obesity (%) * | Prevalence of Abdominal Obesity (%) * |
| --- | --- | --- | --- |
| Age Group (in Years) | | | |
| 18–24 | 170,357 (24.47) | 4.76 | 42.98 |
| 25–34 | 229,360 (32.67) | 11.59 | 54.32 |
| 35–44 | 197,342 (28.21) | 16.23 | 59.80 |
| 45–54 | 101,227 (14.65) | 17.15 | 63.85 |

**Table 1.** *Cont.*

| Variable | Unweighted Frequency n (%) | Prevalence of Obesity (%) * | Prevalence of Abdominal Obesity (%) * |
|---|---|---|---|
| Gender | | | |
| Male | 88,019 (12.43) | 11.56 | 54.78 |
| Female | 610,267 (87.57) | 16.31 | 60.84 |
| Highest Educational Status | | | |
| No formal schooling | 160,276 (22.44) | 7.98 | 54.37 |
| Up to primary | 93,526 (13.55) | 11.75 | 56.63 |
| Up to secondary | 335,078 (46.95) | 16.39 | 58.93 |
| College and higher | 109,406 (17.07) | 20.71 | 61.30 |
| Marital Status | | | |
| Never married | 131,465 (17.47) | 10.50 | 57.55 |
| Currently married | 536,888 (78.20) | 14.11 | 58.48 |
| Others [a] | 29,933 (4.33) | 11.12 | 54.15 |
| Household Wealth Status | | | |
| Poorest | 140,641 (18.02) | 4.59 | 53.07 |
| Poorer | 153,614 (20.04) | 8.00 | 55.18 |
| Middle | 147,759 (20.91) | 12.41 | 56.72 |
| Richer | 137,007 (21.15) | 17.78 | 59.61 |
| Richest | 119,265 (19.87) | 24.54 | 62.99 |
| Zone of Residence | | | |
| North | 129,441 (13.26) | 15.51 | 63.72 |
| North-East | 101,672 (3.84) | 8.22 | 57.48 |
| Central | 171,091 (24.46) | 11.45 | 56.48 |
| East | 111,567 (22.73) | 9.28 | 64.37 |
| West | 72,405 (14.50) | 13.48 | 46.98 |
| South | 112,110 (21.22) | 21.08 | 56.05 |
| Place of Residence | | | |
| Urban | 173,936 (32.02) | 20.14 | 61.17 |
| Rural | 524,350 (67.98) | 10.65 | 56.04 |
| Current Smoker/Tobacco use | | | |
| No | 602,583 (89.29) | 15.98 | 58.88 |
| Yes | 95,703 (10.71) | 8.28 | 55.23 |
| Current Alcohol Consumption | | | |
| No | 660,919 (96.40) | 14.20 | 57.80 |
| Yes | 37,367 (3.60) | 8.99 | 58.21 |

Unweighted frequency and weighted percentages were reported. [a] Divorced/widowed/separated/don't know; * age and sex standardized prevalence was estimated.

Of the respondents, 13.85% and 57.71% were with obesity and abdominal obesity, respectively. The prevalence of underweight and overweight was 12.04% and 32.69%, respectively. Figure 1 shows the distribution of abdominal obesity according to BMI status. A total of 65.45% of overweight and 73.91% of obese individuals had abdominal obesity. Meanwhile, 52.94% of the participants with normal BMI and 34.46% of the participants with underweight had abdominal obesity.

The prevalence of both obesity and abdominal obesity was higher among the older age group. The prevalence was higher among females (obesity: 16.31%; abdominal obesity: 60.84%) compared to their male counterparts (obesity: 11.56%; abdominal obesity: 54.78%). The prevalence was higher among individuals who had ever been married (currently married/divorced/widowed/separated/don't know) and urban residents. The prevalence of obesity was higher among the participants who had studied up to college level and higher and were residing in the South zone, whereas the prevalence of abdominal obesity was higher among residents from the North zone without formal schooling. Current smokers/tobacco users had a lower prevalence of obesity and abdominal obesity than those who did not smoke/use tobacco.

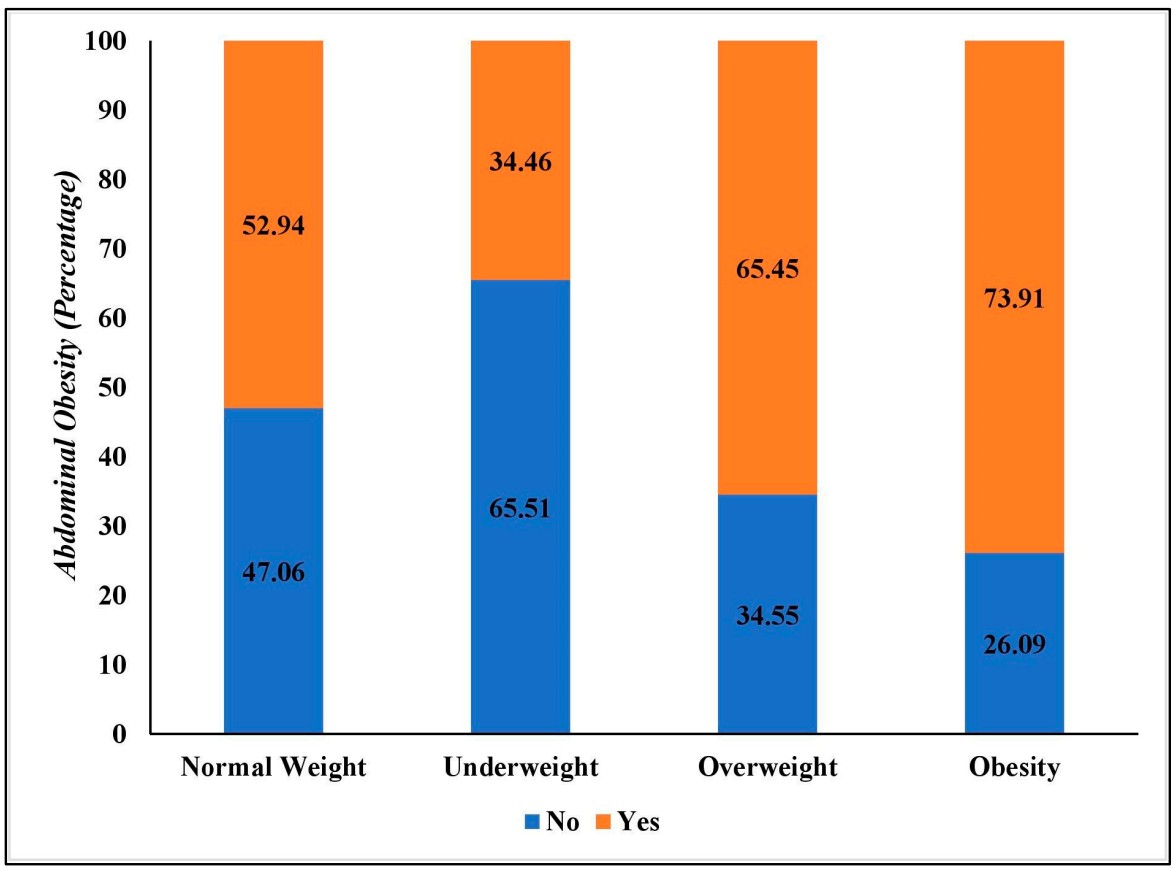

**Figure 1.** Distribution of abdominal obesity according to the BMI status among the participants.

The prevalence of obesity and abdominal obesity by background characteristics among males and female is shown in Supplementary Tables S1 and S2. Table 2 presents the crude and adjusted odds ratio (95% CI) estimates of the factors associated with obesity and abdominal obesity. The odds of both obesity and abdominal obesity increased with age. The odds of having obesity and abdominal obesity were almost 4.29 times (AOR: 4.29; 95% CI: 4.14–4.45, $p < 0.001$) and 2.11 times (AOR: 2.11; 95% CI: 2.06–2.16, $p < 0.001$) among those who were aged 45–54 years compared to those who were aged 18–24 years. Females had 21% higher odds of having obesity (AOR:1.21; 95% CI: 1.17–1.24, $p < 0.001$) and 55% higher odds of having abdominal obesity (AOR: 1.55; 95% CI: 1.52–1.58, $p < 0.001$) compared to their male counterparts. The odds also increased significantly with educational status. Individuals who had ever been married were more likely to suffer from obesity and abdominal obesity than never-married individuals. The odds of obesity were 47% higher among residents of the South zone compared to residents of the North zone (AOR: 1.47; 95% CI: 1.41–1.52, $p < 0.001$). Rural residents had 13% lower odds of abdominal obesity than urban residents (AOR: 0.87; 95% CI: 0.84–0.90, $p < 0.001$). Similarly, rural residents had 27% lower odds of obesity than urban residents (AOR: 0.73; 95% CI: 0.71–0.75, $p < 0.001$). Current smokers had 20% lower odds of obesity (AOR: 0.80; 95% CI: 0.78–0.82, $p < 0.001$) and 9% lower odds of abdominal obesity (AOR: 0.91; 95% CI: 0.90–0.93, $p < 0.001$) compared to those who did not smoke. Current alcohol users had 5% higher odds of abdominal obesity than those who were not current alcohol users (AOR: 1.05; 95% CI: 1.02–1.08, $p < 0.001$).

**Table 2.** Crude and adjusted odds ratio (95% CI) estimates of the factors associated with obesity and abdominal obesity.

| Variable | Obesity | | Abdominal Obesity | |
| --- | --- | --- | --- | --- |
| | COR (95% CI) | AOR (95% CI) | COR (95% CI) | AOR (95% CI) |
| Age Group (in Years) | | | | |
| 18–24 | Ref | Ref | Ref | Ref |
| 25–34 | 2.97 *** (2.89–3.06) | 2.31 *** (2.24–2.39) | 1.49 *** (1.47–1.51) | 1.33 *** (1.31–1.35) |
| 35–44 | 4.99 *** (4.86–5.13) | 3.89 *** (3.76–4.02) | 1.93 *** (1.90–1.96) | 1.72 *** (1.69–1.75) |
| 45–54 | 5.38 *** (5.22–5.54) | 4.29 *** (4.14–4.45) | 2.28 *** (2.24–2.33) | 2.11 *** (2.06–2.16) |
| Gender | | | | |
| Male | Ref | Ref | Ref | Ref |
| Female | 1.24 *** (1.21–1.27) | 1.21 *** (1.17–1.24) | 1.54 *** (1.51–1.57) | 1.55 *** (1.52–1.58) |
| Highest Educational Status | | | | |
| No formal schooling | Ref | Ref | Ref | Ref |
| Up to primary | 1.16 *** (1.13–1.19) | 1.20 *** (1.17–1.24) | 0.91 *** (0.89–0.93) | 1.01 (0.99–1.03) |
| Up to secondary | 1.11 *** (1.08–1.13) | 1.28 *** (1.25–1.31) | 0.80 *** (0.79–0.81) | 1.05 *** (1.03–1.07) |
| College and higher | 0.94 *** (0.91–0.97) | 1.18 *** (1.14–1.22) | 0.71 *** (0.70–0.73) | 1.06 *** (1.04–1.09) |
| Marital Status | | | | |
| Never married | Ref | Ref | Ref | Ref |
| Currently married | 3.74 *** (3.64–3.85) | 1.79 *** (1.73–1.85) | 1.84 *** (1.81–1.87) | 1.32 *** (1.29–1.34) |
| Others [a] | 3.64 *** (3.48–3.81) | 1.62 *** (1.55–1.71) | 1.95 *** (1.89–2.00) | 1.25 *** (1.21–1.29) |
| Household Wealth Status | | | | |
| Poorest | Ref | Ref | Ref | Ref |
| Poorer | 1.90 *** (1.83–1.97) | 1.76 ***(1.70–1.82) | 1.07 (1.06–1.09) | 1.10 (1.08–1.12) |
| Middle | 3.01 *** (2.91–3.12) | 2.57 *** (2.48–2.66) | 1.15 (1.13–1.17) | 1.18 (1.16–1.21) |
| Richer | 4.45 *** (4.30–4.60) | 3.58 *** (3.45–3.72) | 1.25 (1.22–1.28) | 1.28 (1.25–1.31) |
| Richest | 6.70 *** (6.47–6.94) | 5.12 *** (4.92–5.33) | 1.39 (1.35–1.42) | 1.37 (1.34–1.41) |
| Zone of Residence | | | | |
| North | Ref | Ref | Ref | Ref |
| North-East | 0.48 *** (0.46–0.50) | 0.86 *** (0.82–0.90) | 0.74 *** (0.71–0.78) | 0.83 *** (0.79–0.88) |
| Central | 0.58 *** (0.56–0.60) | 0.90 *** (0.87–0.94) | 0.45 *** (0.43–0.47) | 0.50 *** (0.48–0.52) |
| East | 0.48 *** (0.45–0.50) | 0.90 *** (0.86–0.94) | 0.73 *** (0.70–0.77) | 0.83 *** (0.79–0.87) |
| West | 0.72 *** (0.69–0.76) | 0.78 *** (0.75–0.81) | 0.28 *** (0.26–0.29) | 0.27 *** (0.25–0.28) |
| South | 1.45 *** (1.39–1.51) | 1.47 *** (1.41–1.52) | 0.43 *** (0.41–0.45) | 0.40 *** (0.38–0.42) |
| Place of Residence | | | | |
| Urban | Ref | Ref | Ref | Ref |
| Rural | 0.40 *** (0.39–0.41) | 0.73 *** (0.71–0.75) | 0.83 *** (0.80–0.85) | 0.87 *** (0.84–0.90) |
| Current Smoker/Tobacco use | | | | |
| No | Ref | Ref | Ref | Ref |
| Yes | 0.78 *** (0.76–0.80) | 0.80 *** (0.78–0.82) | 0.95 *** (0.93–0.97) | 0.91 *** (0.90–0.93) |
| Current Alcohol Consumption | | | | |
| No | Ref | Ref | Ref | Ref |
| Yes | 0.87 *** (0.83–0.90) | 0.97 (0.93–1.02) | 0.94 *** (0.91–0.96) | 1.05 ** (1.02–1.08) |

**: $p < 0.01$; ***: $p < 0.001$; AOR: adjusted odds ratio; CI: confidence interval; COR: crude odds ratio; [a] divorced/widowed/separated/don't know.

The estimates on the factors associated with obesity and abdominal obesity among males and females are reported in Supplementary Tables S3 and S4, respectively. Similar associated factors were found in both genders; however, there was a difference in the statistical significance among some factors. For example, current alcohol intake was significantly associated with abdominal obesity among females but not among males (AOR: 1.09; 1.04–1.14, $p < 0.001$).

## 4. Discussion

This study determined the prevalence and determinants of obesity and abdominal obesity in India from a nationally representative sample. We found that more than 50% of the respondents had abdominal obesity, while 13.85% of participants had obesity. The prevalence was higher among females compared to their male counterparts. The odds of obesity and abdominal obesity were higher with increasing age, being female, increased educational status, being a resident of the North zone of India (compared to South Zone), having ever been married, being a resident of an urban area, and currently not smok-

ing/using tobacco. Current alcohol intake significantly increased the odds of abdominal obesity only. Gender-segregated analysis revealed mostly similar findings.

The majority of the respondents (57.71%) had abdominal obesity. Previously, a systematic review and meta-analysis conducted by Ahirwar et al. (2019) estimated that around 16.9–36.3% of individuals in India could have abdominal obesity [14]. Other recent primary studies estimated the prevalence between 19% and 71.2% [15,23,24]. This discrepancy in the prevalence might be due to differences in methods, study population, study settings, and operational definitions used for abdominal obesity. The prevalence is also higher than neighboring Bangladesh (42.5%) [13] and China (32.7%) [25]. On the other hand, the prevalence of general obesity was 13.85%, which is slightly higher than the prevalence found in Bangladesh (10.8%) [13].

We found that the prevalence and odds of obesity and abdominal obesity increased with increasing age. This happens because of the tendency of adipose tissue to deviate towards the central abdomen from the periphery [26]. We also found that females were more likely to develop general obesity and abdominal obesity compared to their counterparts, which is consistent with earlier findings [15]. Both age and sex are nonmodifiable risk factors for abdominal obesity. Females and those in the older age group should be targeted for awareness campaigns to limit the load of obesity [27].

There is a positive association between abdominal obesity and educational status, which is aligned with previous literature [13,28]. Highly educated individuals are usually involved in less physical-activity-demanding lifestyles [29]. Being married at any point increased the probability of obesity and abdominal obesity among both genders, which is also like previous studies [13,30,31]. One explanation may be weight gain because of spousal influence on one's eating habits [32]. Interventions addressing obesity should be tailored according to individuals with higher educational status. Interventions should further target Indians who are or have been married to address the high prevalence of abdominal obesity.

The wealth index was found to be significantly associated with obesity and abdominal obesity in the overall sample. In the LMICs, people of higher economic status are at risk of developing overweight, obesity, and abdominal obesity, mainly due to a sedentary lifestyle and the intake of junk foods [13,33]. Public health promotion programs in India aimed at addressing the burden of obesity should focus on individuals irrespective of their wealth status.

We found that urban residents were more likely to have obesity and abdominal obesity compared to their rural counterparts, partly due to the accessibility of junk foods and a lack of moderate-to-vigorous physical activity [34]. Similarly, North and South zone residents had higher odds of developing abdominal obesity and obesity compared to the residents of other zones. Regional differences might be due to cultural differences in lifestyle, diet, and/or physical activity, which warrants further exploration [13].

Current alcohol intake was significantly associated with increased odds of abdominal obesity among females. Alcohol intake is associated with increased adiposity, thereby predisposing the person toward abdominal obesity [34,35]. Being a current smoker was significantly inversely associated with obesity and abdominal obesity among both genders, perhaps due to nicotine's effect on appetite and the facilitation of weight loss [36]. Further exploration is necessary to learn the effect of alcohol intake on abdominal obesity among males.

We found that individuals with overweight and obesity had a higher prevalence of abdominal obesity compared to individuals with a normal BMI. Weight gain due to muscle mass and weight gain due to the accumulation of fat cannot be differentiated by BMI [36,37]. However, it is alarming that 34.46% of individuals with underweight and 52.94% of individuals with normal BMI had abdominal obesity, implying that awareness should be created among all individuals irrespective of their BMI status.

The high burdens of obesity and abdominal obesity in India indicate a simultaneously high burden of noncommunicable diseases (NCDs) in India. It was estimated that NCDs are attributable to 5.87 million deaths annually in India, which is 60% of the total deaths [38]. The rising burden of NCDs is also detrimental to India's economy. A study

by Bloom et al. estimated that between 2012–2030, five NCDs (i.e., diabetes, ischemic heart diseases, cerebrovascular disease, chronic obstructive pulmonary disease, and breast cancer) would lead India to lose USD 2.58 trillion [39]. Obesity and abdominal obesity are risk factors for all the above-mentioned NCDs [5,6]. Obesity and abdominal obesity are also associated with developing insulin resistance, which leads to type-2 diabetes mellitus, and subsequently cardiovascular disease and cerebrovascular disease [40]. Obesity and abdominal obesity also increase the risk of cardiovascular disease and cerebrovascular disease by contributing to atherosclerosis of coronary arteries [41]. Obesity increases the risk of breast cancer. Adiposity is known to increase inflammation, and increases the risk of cancer [42]. Even in individuals with normal BMI, abdominal fat increases the risk of breast cancer [43]. Among individuals suffering from chronic obstructive pulmonary disease, abdominal obesity increases the risk of mortality [44]. The Government of India should integrate abdominal obesity prevention into any NCD prevention programs to achieve the sustainable development goal related to NCDs.

This study has some notable strengths. It utilized a nationally representative sample, making the findings generalizable to the context of India. However, the limitations of the study warrant discussion. First, NHFS-5 is a cross-sectional study; as a result, the temporal relationship between the independent variables and obesity and abdominal obesity could not be established. Second, NHFS-5 did not collect information on some key covariates, including fruit and vegetable intake and physical activity. As a result, those variables could not be included in the final multivariable model. Third, NFHS-5 did not collect information on alcohol intake. Fourth, we used waist–hip ratio as an indicator of abdominal obesity. We did not use other measures of visceral adiposity/abdominal obesity, including waist circumference and 'a body shape index' [45,46]. Finally, we used the BMI-cut off for the Asian population to define obesity, but we used the general waist–hip ratio cut-off (applicable for the global population) to define abdominal obesity. There is no established waist–hip ratio cut-off for the Asian population in general.

**5. Conclusions**

This study identified that more than half of the adult population in India is suffering from abdominal obesity. Although approximately 12% of the population is suffering from obesity, the prevalence of abdominal obesity was high in every BMI subcategory. Older age, female gender, increased educational status and increased wealth index, being married at any point, and being a resident of an urban area all increased the risk of both obesity and abdominal obesity. Being a resident of the North zone and current alcohol intake increased the risk of abdominal obesity. On the other hand, being a resident of the South zone of India increased the probability of obesity. Health promotion programs in India should focus on these selected risk groups to address the high burden of obesity and abdominal obesity.

**Supplementary Materials:** The following supporting information can be downloaded at: https://www.mdpi.com/article/10.3390/epidemiologia4020017/s1, Table S1: Prevalence of obesity and abdominal obesity by background characteristics among males; Table S2: Prevalence of obesity and abdominal obesity by background characteristics among females; Table S3. Crude and adjusted odds ratio (95% CI) estimates of the factors associated with obesity and abdominal obesity among males; Table S4. Crude and adjusted odds ratio (95% CI) estimates of the factors associated with obesity and abdominal obesity among females.

**Author Contributions:** Conceptualization, R.D.G. and M.R.H.; methodology, R.D.G., N.T. and M.R.H.; software, R.D.G.; validation, R.D.G., N.T., N.S., S.S.H., E.H.A. and M.R.H.; formal analysis, R.D.G. and M.R.H.; investigation, R.D.G., N.T., N.S., S.S.H., E.H.A. and M.R.H.; resources, R.D.G.; data curation, R.D.G., N.T., N.S., S.S.H., E.H.A. and M.R.H.; writing—original draft preparation, R.D.G. and N.T.; writing—review and editing, R.D.G., N.T., N.S., S.S.H., E.H.A. and M.R.H.; visualization, R.D.G., N.T., N.S., S.S.H., E.H.A. and M.R.H.; supervision, E.H.A. and M.R.H.; project administration, R.D.G. All authors have read and agreed to the published version of the manuscript.

**Funding:** This research received no external funding.

**Institutional Review Board Statement:** The study protocol of NFHS-5 was approved by the institutional review board of the International Institute for Population Sciences (IIPS) and ICF. The U.S. Centers for Disease Control and Prevention (CDC) also reviewed the protocol. We received approval from DHS program for using the NFHS-5 data for this study in August 2022.

**Informed Consent Statement:** Informed consent was obtained from all subjects involved in the study.

**Data Availability Statement:** The data can be accessed from the DHS program following proper procedure through this url: https://dhsprogram.com/Data/. We accessed the data on 15 August 2022.

**Conflicts of Interest:** The authors declare no conflict of interest.

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
