# Peer review of "Obesity and Abdominal Obesity in Indian Population: Findings from a Nationally Representative Study of 698,286 Participants"

_epidemiologia, doi:10.3390/epidemiologia4020017_

Round 1
Reviewer 1 Report
This nationwide study describes the prevalence of general obesity and abdominal obesity in the Indian population, and found that the prevalence of abdominal obesity is higher in Indian adults. It also identified the risk groups for general obesity and abdominal obesity, providing evidence for further research and policies. I have some suggestions for the authors to consider.
As a nationwide study, a map displaying the obesity prevalence in different zones of residence may help readers better understand the study.
Please carefully check the contents of the tables, as there may be errors in the asterisks in Table 2 - Abdominal Obesity-COR.
The authors indicated that being a resident of the South zone of India has increased the odds of general obesity, but this conclusion is only based on the comparison with the North zone, and the comparison of the South zone with other regions is not shown, so the evidence for this conclusion is not sufficient.
In the methods section, please clarify the criteria for stratifying the wealth index into poorest, poorer, middle, richer, and richest categories, and specify the income or wealth ranges associated with each level. Although these stratification methods may have been mentioned in other studies, it is still recommended to include them in this article.
Author Response
Reviewer 1:
This nationwide study describes the prevalence of general obesity and abdominal obesity in the Indian population, and found that the prevalence of abdominal obesity is higher in Indian adults. It also identified the risk groups for general obesity and abdominal obesity, providing evidence for further research and policies. I have some suggestions for the authors to consider.
As a nationwide study, a map displaying the obesity prevalence in different zones of residence may help readers better understand the study.
Response: Thanks! This map is not within the scope of this study.
Please carefully check the contents of the tables, as there may be errors in the asterisks in Table 2 - Abdominal Obesity-COR.
Response: Thanks! We have revised the estimates
The authors indicated that being a resident of the South zone of India has increased the odds of general obesity, but this conclusion is only based on the comparison with the North zone, and the comparison of the South zone with other regions is not shown, so the evidence for this conclusion is not sufficient.
Response: Thanks! We have mentioned it in the bracket that this comparison was made with South Zone.
In the methods section, please clarify the criteria for stratifying the wealth index into poorest, poorer, middle, richer, and richest categories, and specify the income or wealth ranges associated with each level. Although these stratification methods may have been mentioned in other studies, it is still recommended to include them in this article.
Response: Thanks! There is no income related information in NDHS-5. DHS program calculate wealth index and categorize into five quintiles: poorest, poorer, middle, richer, and richest. This categorization allowed for the classification of households into different wealth categories based on their relative wealth status compared to other households in the survey. NDHS-5 already provided the wealth quintiles data in the dataset.
Reviewer 2 Report
Review of manuscript
Epidemiological study that aimed to estimate the prevalence and correlates of abdominal obesity (aged ≥15 years) and compare it with the prevalence and factors related to obesity using a nationally representative sample of the Indian population obtained from the National Family Health Survey 2019-21.
Below make my considerations in each part of manuscript
Abstract
Need a moderate English improvement
Nationally? Maybe the correct form is national. Please, check in all manuscript
Determine – no deter-mine
Introduction
Line 40 - adipose (fat) tissue is … it is not necessary to include fat between parentheses. Please, check in all paper.
Line 43- In the scientific literature is not useful to mention general or generalized obesity, suggest writing obesity and abdominal obesity.
Line 51 – we can use word that mean stigmatization. So, please substitute in all manuscript OBESE for WITH OBESITY
Line 52-54 – This phrase is understandable. “Yet, it remains one of the least addressed 52 health issues and is a major public health concern in upper-both upper income and low- 53 and-middle-income countries (LMICs).”
Line 57- 62 – This part of manuscript can be improved.” With recent economic progress and urbanization, traditional 57 healthy ways of eating have been substituted by Western-influenced diets rich in saturated fats, refined sugar, and preservatives. Traditional eating of white rice and lack of 59 physical activity may also contribute deposition of abdominal fat among South Asian people [12].
Line 61 – This affirmation is not completely correct because we can find some data about prevalence of abdominal obesity in South Asian region. However, there is a lack of literature on the prevalence of abdominal obesity in 61 the South Asian region [13].”
Methodology
Line 73 – 75 – Is very important to mention on this section information about sample size calculation and briefly recruitment of the sample.
“A secondary analysis of the NFHS-5 2019-21 was conducted. NFHS-5 was a nation- ally representative survey implemented in India. The detailed report of methodology, sample size calculation, and preliminary findings were published elsewhere [16].”
Line 85 – Abdominal obesity was defined as a waist-hip 85 ratio of >0.90 for males and >0.85 for females, but nowadays we have many other indexes that can verify this measure more precisely. Maybe is a limitation of the study.
Line 98 – can explain better this evaluation, please.
“The wealth index was computed using principal component analysis [16].”
In methodology miss the evaluation of physical activity. This is very important and completely related to the increase of abdominal adiposity.
The authors have information about quantity of alcohol intake?
Results
Withdraw the lines of tables (main manuscript and supplementary)
Figure 1 needs to insert nomenclature in vertical axis.
Discussion
This section could be more interesting with some metabolic explanation about the results, even being an epidemiological study.
Author Response
Reviewer 2:
Review of manuscript
Epidemiological study that aimed to estimate the prevalence and correlates of abdominal obesity (aged ≥15 years) and compare it with the prevalence and factors related to obesity using a nationally representative sample of the Indian population obtained from the National Family Health Survey 2019-21.
Below make my considerations in each part of manuscript
Response: Thank you very much. In this study we included participants aged 15-54 years, as only 40 participants were found above the age of 54 years (among 777,889 participants) and we decided to drop them.
Abstract
Need a moderate English improvement
Response: Thanks we have revised the abstract.
Nationally? Maybe the correct form is national. Please, check in all manuscript
Response: The correct form is "nationally representative sample." This phrase refers to a sample that has been selected in a way that ensures it accurately reflects the demographic and geographic characteristics of the entire nation. "National representative sample" could be interpreted as a sample that represents a single national entity or organization, rather than a broader population.
Determine – no deter-mine
Response: Thanks! We have corrected it.
Introduction
Line 40 - adipose (fat) tissue is … it is not necessary to include fat between parentheses. Please, check in all paper.
Response: Thanks. We have edited this throughout the manuscript.
Line 43- In the scientific literature is not useful to mention general or generalized obesity, suggest writing obesity and abdominal obesity.
Response: Thanks. We have edited this throughout the manuscript.
Line 51 – we can use word that mean stigmatization. So, please substitute in all manuscript OBESE for WITH OBESITY
Response: Thanks. We have edited this throughout the manuscript.
Line 52-54 – This phrase is understandable. “Yet, it remains one of the least addressed 52 health issues and is a major public health concern in upper-both upper income and low- 53 and-middle-income countries (LMICs).”
Response: Thanks! We have edited it accordingly: “Yet, it remains one of the least addressed health issues and is a major public health concern in both upper- income and low-and-middle-income countries (LMICs).”
Line 57- 62 – This part of manuscript can be improved.” With recent economic progress and urbanization, traditional 57 healthy ways of eating have been substituted by Western-influenced diets rich in saturated fats, refined sugar, and preservatives. Traditional eating of white rice and lack of 59 physical activity may also contribute deposition of abdominal fat among South Asian people [12].
Response: I have revised it accordingly: “ With recent economic progress and urbanization, traditional healthy diets have been substituted by Western-influenced diets rich in saturated fats, refined sugar, and preservatives. Excessive consumption of white rice and lack of physical activity may also contribute deposition of abdominal fat among South Asian people [12].”
Line 61 – This affirmation is not completely correct because we can find some data about prevalence of abdominal obesity in South Asian region. However, there is a lack of literature on the prevalence of abdominal obesity in 61 the South Asian region [13].”
Response: Thanks! We have removed this sentence.
Methodology
Line 73 – 75 – Is very important to mention on this section information about sample size calculation and briefly recruitment of the sample.
Response: Thanks! We have added the following section: “Stratified sampling strategy was followed for data collection. Data was collected in two stages. First, primary sampling units (PSUs) were selected from the rural area and census enumeration blocks (CEBs) were selected from the urban area based on probability proportionate to size. At the second stage, an equal number of households (n=22) were selected randomly from each PSU. From the selected households, all the eligible men and women were included for interview [16-17]. In this study we included participants aged 15-54 years, as only 40 participants were found above the age of 54 years (among 777,889 participants).”
The detailed sample size calculation for the original study has been mentioned in the original reports. This study included 15-54 years participants and conducted a complete case analysis.
“A secondary analysis of the NFHS-5 2019-21 was conducted. NFHS-5 was a nation- ally representative survey implemented in India. The detailed report of methodology, sample size calculation, and preliminary findings were published elsewhere [16].”
Response: Thanks! We have corrected it.
Line 85 – Abdominal obesity was defined as a waist-hip 85 ratio of >0.90 for males and >0.85 for females, but nowadays we have many other indexes that can verify this measure more precisely. Maybe is a limitation of the study.
Response: We humbly disagree with the comment. Both waist-to-hip ratio (WHR) and waist circumference (WC) are commonly used measures of abdominal obesity, but each measure has its strengths and limitations.WHR is calculated by dividing the waist circumference by the hip circumference. This measure is often used because it takes into account the distribution of fat in the body and may be a better predictor of cardiovascular disease risk than other measures of body fatness. A WHR of greater than 0.85 in women or 0.90 in men is generally considered to indicate abdominal obesity. On the other hand, WC is a simpler measure that is easier to obtain and may be more practical in some settings. A WC of greater than 88 cm (35 inches) in women or 102 cm (40 inches) in men is generally considered to indicate abdominal obesity. It's important to note that neither measure is perfect, and they should be used in conjunction with other measures of body composition and cardiovascular disease risk factors.
Line 98 – can explain better this evaluation, please.
“The wealth index was computed using principal component analysis [16].”
Response: Thanks! We have revised the statement as follows: “This information was used to compute a wealth index using principal component analysis.”
In methodology miss the evaluation of physical activity. This is very important and completely related to the increase of abdominal adiposity.
Response: Thanks! We have mentioned this in the limitation: “Second, NHFS-5 did not collect information on some key covariates including fruit and vegetable intake and physical activity. As a result, those variables could not be included in the final multivariable model.”
The authors have information about quantity of alcohol intake?
Response: Thanks! We do not have information about quantity of alcohol intake. In the limitation section, we have added the following “Third, NFHS-5 did not collect the information on the quantity of alcohol intake.”
Results
Withdraw the lines of tables (main manuscript and supplementary)
Response: Thanks! The tables are formatted according to epidemiologia guideline.
Figure 1 needs to insert nomenclature in vertical axis.
Response: Thanks we have edited it.
Discussion
This section could be more interesting with some metabolic explanation about the results, even being an epidemiological study.
Response: We have already added both social and metabolic explanation about the results.
Round 2
Reviewer 2 Report
Review of manuscript (round 2)
Dear authors, thank you for all replies. I need to add my concerns about the revised manuscript.
- Line 85 – Abdominal obesity was defined as a waist-hip ratio of >0.90 for males and >0.85 for females, but nowadays we have many other indexes that can verify this measure more precisely. Maybe is a limitation of the study.
Response: We humbly disagree with the comment. Both waist-to-hip ratio (WHR) and waist circumference (WC) are commonly used measures of abdominal obesity, but each measure has its strengths and limitations.WHR is calculated by dividing the waist circumference by the hip circumference. This measure is often used because it takes into account the distribution of fat in the body and may be a better predictor of cardiovascular disease risk than other measures of body fatness. A WHR of greater than 0.85 in women or 0.90 in men is generally considered to indicate abdominal obesity. On the other hand, WC is a simpler measure that is easier to obtain and may be more practical in some settings. A WC of greater than 88 cm (35 inches) in women or 102 cm (40 inches) in men is generally considered to indicate abdominal obesity. It's important to note that neither measure is perfect, and they should be used in conjunction with other measures of body composition and cardiovascular disease risk factors.
Reviewer (round 2): Dear authors, if you research more manuscripts about abdominal obesity, how we can evaluate it, you can find many others index with more sensitivity, specificity, and accuracy. So, I reinforce that this can be a limitation of the study.
- Line 98 – can explain better this evaluation, please.
“The wealth index was computed using principal component analysis [16].”
Response: Thanks! We have revised the statement as follows: “This information was used to compute a wealth index using principal component analysis.”
Reviewer (round 2): Dear authors, this part f the review continues without my understanding. It is necessary to give a little bit more information.
- Figure 1 needs to insert nomenclature in vertical axis.
Response: Thanks we have edited it.
Reviewer (round 2): Dear authors, nomenclature in vertical axis is abdominal obesity (percentage).
Discussion
- This section could be more interesting with some metabolic explanation about the results, even being an epidemiological study.
Response: We have already added both social and metabolic explanation about the results.
Reviewer (round 2): Dear authors, I reinforce that the discussion could be more interesting. Miss more metabolic discussion, that becomes the discussion strengthen.
Conclusion
Reviewer (round 2): This part of conclusion is not possible to be mention, because the authors do not evaluate the alcohol intake. “Being a resident of North zone and current alcohol intake increased the risk of abdominal obesity”
Author Response
Dear authors, thank you for all replies. I need to add my concerns about the revised manuscript.
- Line 85 – Abdominal obesity was defined as a waist-hip ratio of >0.90 for males and >0.85 for females, but nowadays we have many other indexes that can verify this measure more precisely. Maybe is a limitation of the study.
Response: We humbly disagree with the comment. Both waist-to-hip ratio (WHR) and waist circumference (WC) are commonly used measures of abdominal obesity, but each measure has its strengths and limitations.WHR is calculated by dividing the waist circumference by the hip circumference. This measure is often used because it takes into account the distribution of fat in the body and may be a better predictor of cardiovascular disease risk than other measures of body fatness. A WHR of greater than 0.85 in women or 0.90 in men is generally considered to indicate abdominal obesity. On the other hand, WC is a simpler measure that is easier to obtain and may be more practical in some settings. A WC of greater than 88 cm (35 inches) in women or 102 cm (40 inches) in men is generally considered to indicate abdominal obesity. It's important to note that neither measure is perfect, and they should be used in conjunction with other measures of body composition and cardiovascular disease risk factors.
Reviewer (round 2): Dear authors, if you research more manuscripts about abdominal obesity, how we can evaluate it, you can find many others index with more sensitivity, specificity, and accuracy. So, I reinforce that this can be a limitation of the study.
Response (round 2): Thanks! We have added this as a limitation: “Finally, we used waist-hip ratio as an indicator of abdominal obesity. We did not use other measures of visceral adiposity/ abdominal obesity including waist circumference and ‘a body shape index’ [45,46].”
- Line 98 – can explain better this evaluation, please.
“The wealth index was computed using principal component analysis [16].”
Response: Thanks! We have revised the statement as follows: “This information was used to compute a wealth index using principal component analysis.”
Reviewer (round 2): Dear authors, this part f the review continues without my understanding. It is necessary to give a little bit more information.
Response (round 2): Thanks! Please look at the revised manuscript: “Since DHS program did not collect information on the household income, wealth index which is a composite measure of a household's cumulative living standard was used to represent the economic condition of the hosueholds. This information was used to compute a wealth index using principal component analysis [16]. After computing the wealth index, it was categorized into poorest, poorer, middle, richer, and richest. Marital status was categorized into never married, currently married, and others (divorced/widowed/separated/don’t know). The residence zone was categorized according to the current administrative zones of India: North, North-East, Central, East, West, and South [19].”
- Figure 1 needs to insert nomenclature in vertical axis.
Response: Thanks we have edited it.
Reviewer (round 2): Dear authors, nomenclature in vertical axis is abdominal obesity (percentage).
Response (round 2): Thanks! We have edited it.
Discussion
- This section could be more interesting with some metabolic explanation about the results, even being an epidemiological study.
Response: We have already added both social and metabolic explanation about the results.
Reviewer (round 2): Dear authors, I reinforce that the discussion could be more interesting. Miss more metabolic discussion, that becomes the discussion strengthen.
Response (round 2): Thanks! We have added the following metabolic discussion: “Obesity and abdominal obesity are also associated with developing insulin resistance which leads to type-2 diabetes mellitus and subsequently cardiovascular disease and cerebrovascular disease [40]. Obesity and abdominal obesity also increase the risk of cardiovascular disease and cerebrovascular disease by contributing to atherosclerosis of coronary arteries [41]. Obesity increases the risk of breast cancer. Adiposity is known to increase inflammation and increases the risk of cancer [42]. Even in individual with normal BMI, abdominal fat increases the risk of breast cancer [43]. Among the individuals suffering from chronic obstructive pulmonary disease, abdominal obesity in-creases the risk of mortality [44].”
Conclusion
Reviewer (round 2): This part of conclusion is not possible to be mention, because the authors do not evaluate the alcohol intake. “Being a resident of North zone and current alcohol intake increased the risk of abdominal obesity”
Response (round 2): Thank you for this comment. NFHS-5 did not collect the information on the quantity of alcohol intake but it reported current alcohol intake. We have mentioned in the limitation section that, “Third, NFHS-5 did not collect the information on the quantity of alcohol intake.” We did evaluate current alcohol intake.
